

# Bayesian network model structure based on binary evolutionary algorithm

Yongna Yao

School of Information and Electronic Engineering, Shangqiu Institute of Technology, Shangqiu, China

## ABSTRACT

With the continuous development of new technologies, the scale of training data is also expanding. Machine learning algorithms are gradually beginning to be studied and applied in places where the scale of data is relatively large. Because the current structure of learning algorithms only focus on the identification of dependencies and ignores the direction of dependencies, it causes multiple labeled samples not to identify categories. Multiple labels need to be classified using techniques such as machine learning and then applied to solve the problem. In the environment of more training data, it is very meaningful to explore the structure extension to identify the dependencies between attributes and take into account the direction of dependencies. In this article, Bayesian network structure learning, analysis of the shortcomings of traditional algorithms, and binary evolutionary algorithm are applied to the randomized algorithm to generate the initial population. In the optimization process of the algorithm, it uses a Bayesian network to do a local search and uses a depth-first algorithm to break the loop. Finally, it finds a higher score for the network structure. In the simulation experiment, the classic data sets, ALARM and INSURANCE, are introduced to verify the effectiveness of the algorithm. Compared with NOTEARS and the Expectation-Maximization (EM) algorithm, the weight evaluation index of this article was 4.5% and 7.3% better than other schemes. The clustering effect was improved by 13.5% and 15.2%. The smallest error and the highest accuracy are also better than other schemes. The discussion of Bayesian reasoning in this article has very important theoretical and practical significance. This article further improves the Bayesian network structure and optimizes the performance of the classifier, which plays a very important role in promoting the expansion of the network structure and provides innovative thinking.

## INTRODUCTION

Although the development of network technology has brought about information sharing, it has also brought the world into the stage of information explosion. Most of the data collected are incomplete and there are missing values. The Bayesian network structure learning with incomplete data is more complex than Bayesian network structure learning with complete data, which costs more computing resources. It is more challenging and has more practical significance (*He et al., 2020*). The Bayesian network can be applied in data mining, and it has been applied in many fields. The reliable Bayesian network structure is

Corresponding author
Yongna Yao,
1350007356@sqgxy.edu.cn

the main consideration of its application. In the face of a large number of data, it is too laborious to rely solely on artificial experience to construct the Bayesian network structure. Thus, the Bayesian network structure learning algorithm is widely used to construct the network structure. At present, the meta-heuristic algorithm with high popularity is adopted, and the scoring search method is adopted too. By adding an additional stage in the exploration and development stage of the algorithm, the premature local optimization of the algorithm is avoided. The local search ability is increased (*Bellmann & Schwenker, 2020*).

Classification is a key problem in artificial intelligence and machine learning, and it is very important to build a good classifier. Since the naive Bayesian network (NB) has achieved great success in classification tasks, its good classification performance and few computing resources have made more and more researchers begin to pay attention to Bayesian network classifier (BNC) (*Li et al., 2017*). However, the independence assumption of NB is too idealistic and rarely holds in practical applications. Therefore, improving the independence assumption of NB is the key issue to improving the classification performance of NB. Among the various methods to improve NB, the average first-order dependency estimator (AODE) has achieved excellent classification performance by virtue of structural expansion and ensemble learning strategies, although the independence assumption of each of its sub-classifiers (super-parent attribute first-order dependency estimator, SPODE) is rarely true in practical applications (*Singer, Anuar & Ben-Gal, 2020*). As the scale of data continues to expand, robust classifiers with high expressive power and low bias are urgently needed.

Typical constraint-based algorithms include the causal induction algorithm and the three-stage independence analysis algorithm. These algorithms all adopt statistical tests (*Vargas, Gutiérrez & Hervás-Martínez, 2020*). For each pair of variables, there may be a separator variable that makes the pair conditionally independent. The Peter-Clerk (PC) algorithm limits the scope of the separator variable to the set of variables adjacent to the pair. The main idea of the recursive algorithm in *Fulford et al. (2020)* is to first decompose a legally undirected independent graph into a series of subgraphs that may be independently learned. After the structure of the subgraph is learned, the small subgraphs are combined step by step to restore the large graph. Finally, the criterion proposed in *Yu & Smith (2021)* is adopted to determine the direction of as many edges as possible. *Castelletti & Mascaro (2021)* proposes to create and expand a training set based on a selected instance. The training instances can be cloned according to the similarity of the test instances to achieve the purpose of expanding the training set and learning a Bayesian network classifier (BNC). Starting from the test cases themselves, *Ignavier, Ghassami & Zhang (2020)* proposes to use local mutual information (local MI) and local conditional mutual information (local CMI) to dynamically identify the mutual dependencies and conditional dependencies between attribute values. *Halbersberg, Wienreb & Lerner (2020)* proposes an enhanced classifier with external features of label data, which uses a specific method to select the external features of model design with the help of multi-tree basic structure according to the external feature space of low-dimensional label related data in the learning stage. *Liu et al. (2019)* proposes a multi-label extrinsic feature selection method based on information association,

which extends the associated information to multiple pieces of relevant information and combines the basic criteria of multi-block information for evaluation. The extrinsic features do not consider the relevance of labels or use the basic criteria of minimizing multi-block information for evaluation. In the above research results, only the correlation between external characteristics and labels is emphasized. They seldom consider the correlation between tags and cannot effectively reduce the sensitivity to predictable tags. To express the correlation between tags and the reliability of external features of data, further research is needed.

In this article, we improve the traditional algorithm based on the Bayesian network to calculate the causal effect between variables in the case of single intervention and joint intervention. The algorithm introduces the network model to obtain the external feature attributes, and combines the sinusoidal function linear distance to update the Bayesian connection in real time, which achieves good results. In the continuous search space, multivariate individuals update their positions by adding the value of the position vector to the value of the step vector. The likelihood function is applied to add new edges to SPODE, which improves the expressive power of the model on the one hand. On the other hand, it improves the accuracy of estimating causal effects between variables by inheriting the ensemble learning strategy of the Averaged One-Dependent Estimator (AODE) model. Experiments indicate that it is of great significance to learn the precise network topology or the dependencies between attributes for building a good Bayesian network classifier.

The main contents of the article are as follows:

Section 1: According to the Bayesian network based on the binary evolutionary algorithm used in this article, the research results of the network related to machine learning are described.

Section 2: It mainly introduces the basic knowledge of the Bayesian network model and explains the related theory of the two-tuple classifier.

Section 3: The machine learning algorithm is applied to introduce binary clustering and penalty function optimization decision rules in detail.

Section 4: The structure of the Bayesian network is constructed according to the optimization learning algorithm, and the operation process and mathematical principle of the model are introduced.

Section 5: Take the ATODE model as an example. It shows the whole process of the ESPODEI model, which is extended from the corresponding SPODEI through the identification of the dependency relationship and dependency direction between attributes.

Section 6: The experimental analysis compares the NOTEARS algorithm with the EM algorithm and compares the differences between the machine learning methods in the classification accuracy, which is to evaluate the algorithm proposed in this article.

Section 7: Summarize the machine learning algorithms proposed in this article and the experimental results, and elaborate on the main contributions of this article. In view of this article in the research process insufficiency and the difficulty, it proposes further research direction and the possibility.

The main innovations of this article are:

(1) An improved binary optimization algorithm is applied to the Bayesian network structure learning to solve the problem of a directed acyclic graph with poorly fitting data.

(2) It improves the transition parameters of the control stage, strengthens the exploration stage, and makes the algorithm nonlinear and smooth transition to the development stage.

(3) It accepts that non-optimal solution with a certain probability to avoid advancing to the current global optimal direction too early in the exploration stage so as to explore more areas.

## RELATED WORK

### The Bayesian network model

It is not feasible to apply the joint distribution directly for multivariate probabilistic inference, and the computational complexity is super-exponential with the variables. The Bayesian network decomposes the joint probability by using conditional independence between variables, which reduces the number of parameters and is a powerful tool for probabilistic reasoning. The Bayesian network inference includes the posterior probability problem, the maximum posterior probability problem (Maximum a Posterior estimate, MAP), and the maximum probable explanation problem (most probable explanation, MPE) (*Chiribella et al., 2010*). Generally speaking, probabilistic reasoning refers to the problem of the posterior probability. At present, two heuristic algorithms, the maximum potential search, and the minimum missing edge search, are commonly used. *Shafer & Shenoy (1990)* proposed the clique tree propagation algorithm, which uses step sharing to save at least half of the time to complete the posterior inference. In essence, the clique tree propagation algorithm and the variable elimination method are the same, and both are used for exact inference (*Henckel, Perkovi & Maathuis, 2022*). When the network nodes are numerous and dense, the computational complexity is high. Thus, the approximate reasoning algorithm is usually used. The typical approximate reasoning methods include the random sampling method, including repetitive sampling and Markov Chain Monte Carlo (MCMC) sampling; variable distribution method, including naive average length method and loop propagation method; the model simplification method, including removing variables with less influence and reducing the state space of variables; including search-based algorithm, top-N algorithm, deterministic algorithm, MPE approximation algorithm based on an ant colony, etc. (*Sood, 2019*).

It uses an adjacency matrix or adjacency list to represent a directed acyclic graph. In the process of finding the final solution, every Bayesian network is a candidate, and the Bayesian network is expressed as:

$$\lambda = (\alpha, \beta) \tag{1}$$

where $\lambda$ is the number of nodes in the network. The topological structure of a Bayesian network can be expressed as the following matrix form when it means that there is a directed edge between node $\alpha$ and node $\beta$. When the direction is $\alpha$, it means that there

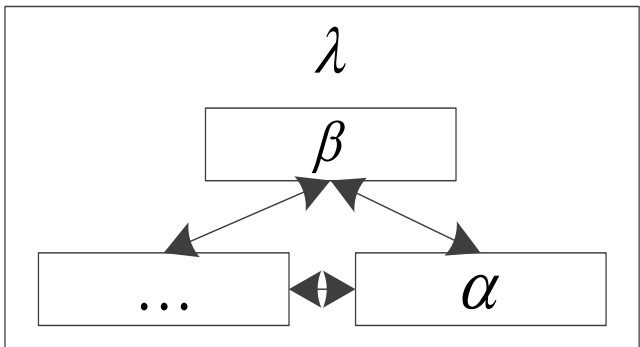

**Figure 1** **Topology of the traditional Bayesian network.**

is no edge between node $\alpha$ and node $\beta$ (*Saengkyongam & Silva, 2020*). The topology is displayed in Fig. 1.

Bayesian networks adopt an adjacency matrix to represent the topological structure of Bayesian networks, which can transform the changes in the abstract topological structure into the changes in the elements of the matrix. The optimization algorithm is to optimize the topology of the Bayesian network, in fact, it is to optimize its adjacency matrix (*Khalifa Othman et al., 2022*).

## Two-tuple classifier

A two-tuple, $\alpha$ represents a network topology with a set of attributes $\gamma = (\gamma_1, \gamma_2, \ldots, \gamma_n)$ and class variables $\beta$ (*Xiangyuan et al., 2021*). $\alpha$ is a DAG (Directed acyclic graph) as shown in Fig. 2.

In a directed acyclic graph, the vertices in the graph represent any attribute, and the edges in the graph represent the probabilistic dependencies between attributes.

The $\beta$ in the two-tuple refers to a set of parameters used to quantify the entire Bayesian network (*Liu et al., 2021*). Bayesian network classifier is a special type of Bayesian network in classification problems. Given an unlabeled instance $\gamma = (\gamma_1, \gamma_2, \ldots, \gamma_n)$, the Bayesian network classifier $\lambda$ assigns a class label $\varphi$ to $\gamma$ with the maximum posterior probability by Eq. (2) (*Duan et al., 2020*).

$$\Delta\varphi_\lambda = \mathrm{argmin}\rho_{\varphi'} \sum_{i=1}^{n} \rho \, |\alpha_i \beta_i| \tag{2}$$

where, $\rho$ represents the set of variables other than class variable for attribute in.

## Binary improved algorithm

The constraint-based structure learning algorithm is to find out the conditional independence relationship between nodes. If the nodes are conditionally independent, there is no edge, otherwise, there is an edge between nodes, such as the PC (Peter-Clerk) algorithm (*Srivastava, Chockalingam & Aluru, 2020*). The structure learning algorithm based on the scoring search is to list all possible Bayesian network structures including all nodes. Then, score each network structure with a specified scoring method. Finally,

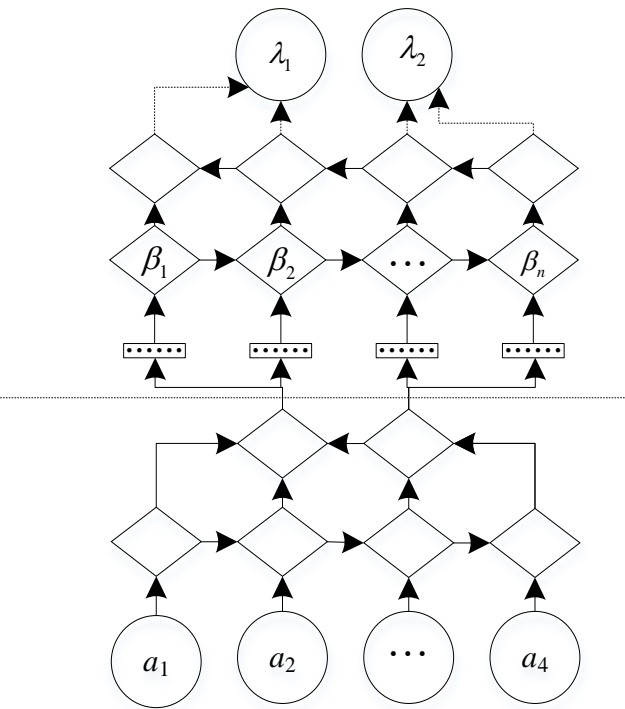

**Figure 2  Two-tuple classifier.**

construct the Bayesian network structure according to a certain search strategy and scoring criteria, such as the GES (Optimal Structure Identification With Greedy Search) algorithm. Mixture-based structure learning algorithm refers to the use of a conditional independence test to reduce the complexity of the search space. Then, score search to find the optimal Bayesian network, such as MMHC algorithm (*Wang et al., 2019*).

In this article, a binary adaptive Lasso algorithm called NS-DIST is proposed, in which the first stage performs neighborhood selection. Then, the second stage estimates the DAG by a discrete improved tabu search (DIST) algorithm based on steepest descent, loop elimination, and tabu list in the selected neighborhood.

## Binary clustering

Inspired by static group behavior and dynamic group behavior, the binary will be divided into many subgroups and operated on different regions, which corresponds to the spatial exploration stage. Binaries will also cluster together to form a large group of operations along a direction, which corresponds to the stage of local mining. Binary behavior follows the principles of grouping, separating, pairing, staying away from attacks, and being attracted to the source of the target. Each binary in the population represents a solution to the exploration space (*Zhang, Petitjean & Buntine, 2020*). Binary group movements are determined by a number of basic operations: separation, alignment, cohesion, being attracted to the target and staying away from the attack (*Yosuf et al., 2022*). Wherein the target node is the individual position of the optimal fitness value explored by the population

in history. The attacking node is the individual position of the worst fitness value explored by the population in history. Separation refers to the static collision that occurs in order to avoid separation from the individual next to it. Formation means that the speed of an individual matches the speed of other individuals around him. Clustering refers to the tendency of individuals to go to the center of the neighborhood (*Nouri-Moghaddam, Ghazanfari & Fathian, 2021*).

In the process of binary clustering, as the domain radius of other individuals around an individual becomes larger, they often cluster into small groups, cluster within the domain radius, or expand the range of activities to stay away from attacks (*Jumelet et al., 2021*).

The behavior of the binary is influenced by the combination of these five modes. To update the position and simulated movement of a binary in the search space, a step vector and a position vector are introduced (*Atoui et al., 2019*).

The step vector represents the direction of binary movement, which is defined as:

$$\Delta f_t = \left[ qW_i + yR_i + pU_i + gK_i + vH_i \right] \varpi \, \Delta \phi_t \tag{3}$$

where $\Delta f_t$ is the mathematical modeling of the segregation behavior of the $i$ th individual in the subpopulation.

$y$ is the mathematical modeling of the formation behavior of the $i$ th individual in the subgroup.

$P$ is the target weight, which is the mathematical modeling of the behavior of the $i$ th individual attracted by the target.

One of the functions of $v$, which is the exchange function of the behavior of the $i$ th individual in the subgroup away from the attack, is to balance exploration and exploitation.

If the transformation function is unchanged during the iteration process, the probability will be calculated in the same way throughout the optimization process. Changing the transformation function may better balance exploration and development (*Min et al., 2020*).

## Penalty function

Penalized likelihood under the assumption of equal variance of latent variables, the final form of the score function is:

$$\lambda_{(x)} = \sum_{i=1}^{n} \left| \phi_n^k - \phi_m^j \right| \cdot \left( \frac{1}{\mu_i t} \right)^2. \tag{4}$$

In the score function, $\phi_n^k$ is a data matrix of $\alpha \times \beta . \phi_m^j$ is a data vector of the $jth$ variable. $(n, m)$ is a coefficient vector, and $(i, j)$ is a column vector of the $jth$ row of the matrix $\phi$. Where, $\mu_{ij}$ is estimated by canonical lasso regression and the initial penalty parameter is given by Eq. (5):

$$\mu_i t = \max \sum_{i=1}^{n} \left( \frac{1}{|\theta_{ij}|} \right)^2 \left( \frac{1}{|F|^{-1}} \right)^2 \tag{5}$$

where $\theta_{ij}$ is the coefficient vector, and $F$ represents the vector group of the undirected neighborhood matrix.

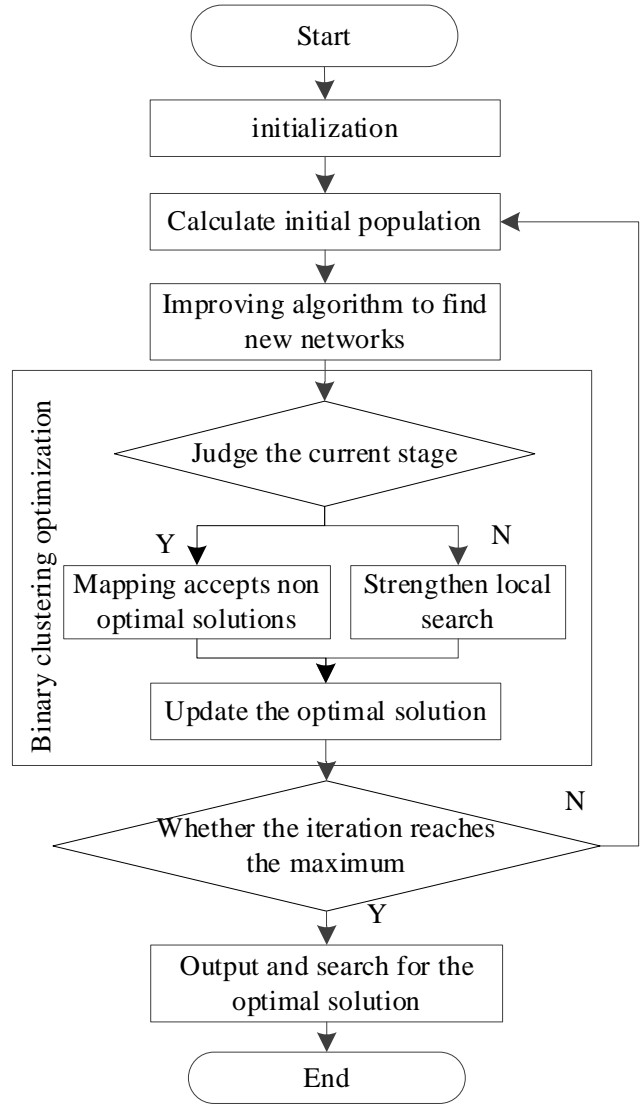

**Figure 3** **Network model operation flow.**

## BAYESIAN NETWORK OPTIMIZATION MODEL

The Bayesian network structure learning algorithm proposed in this article mainly calculates the optimal score, the optimal network structure, and the number of nodes in the initialization stage. In the initial population stage, a plurality of random Bayesian network structures is adopted as initial populations. Next, the optimal network structure in the initial population is found, and its structure and score are given to the initialized optimal network structure and score variables (*Luo, Moffa & Kuipers, 2021*). The improved sine-cosine algorithm is adopted to search each individual in the population to obtain a

new structure. The original individual is replaced when the score is higher than the original individual. The operation process is displayed in Fig. 3.

Pseudo-code for Bayesian network structure learning proposed in this article is as follows:

Input: data set data, node size ns, the maximum number of iterations T.

Output: Bayesian network structure and its score values

1: Randomly generate a plurality of network structures as search operators;

2: Traversal search operator;

3: Update parameters;

4: for i =1:N do

5: Symmetrized adjacency matrix;

6: Binary value calculation;

7: Judge whether the operation forms a loop or not;

8: end{for}

9: for i =1:N do

10: The *if* facilitates the exploration phase *then*.

11: Randomly generate a new network, if the new network is better than the current individual then replace individual

12: Replace the individual

13: end{for}

## CASE STUDY

A case of the ATODE model when $\alpha_1$ acts as a super-parent variable is taken as an example. It shows the whole process of the ESPODEI model which is obtained by extending the corresponding SPODEI after the identification of the dependency relationship and dependency direction between attributes.

(1) $\lambda = (\alpha, \beta)$ and the set of candidate parent variables are $\delta = \phi_t$.

Take the data set mfeat-mor for example, which has six attributes. ATODE needs to find the root node under the condition $\beta_1$. Therefore, it needs to calculate the corresponding $\lambda(\alpha_i | \beta_i), 0 \leq i \leq 6$. The calculation results are displayed in Table 1.

Since the rule for the selection of the root node is to find the attribute with the maximum value of $\lambda(\alpha_i | \beta_i)$, the attribute $\alpha_2$ serves as the root node attribute, and $\alpha_2$ is added to $\delta$. At this point, when the root node $\alpha_2$ is found, it needs to be based on the conditional log-likelihood function to identify conditional dependencies between the remaining attributes $[\alpha_2, \alpha_3, \alpha_4, \alpha_5]$.

(2) The conditional likelihood function values between the remaining attributes need to be calculated, and the calculation results are displayed in Table 2.

In Table 2, INF-[1] is the value that does not need to be calculated because an attribute itself cannot serve as its parent node.

When the root node $\alpha_2$ enters the network structure, it can be seen from Table 2 that the attribute $\lambda(\alpha_i | \beta_i)$ has the largest value. Therefore, the attribute $\lambda(\alpha_i | \beta_i)$ adds $\beta_i$ to its parent attribute set and adds $\alpha_2$ to $\delta$.

Under the condition of the candidate parent variable set $\delta$, the value of the remaining attribute $[\alpha_3, \alpha_4, \alpha_5]$ is the largest, and the ESPODEI is established. The results of data

**Table 1 Data set calculation results (1).**

| Index | Calculation results |
| --- | --- |
| $\alpha_1$ | $0.5718^{-1}$ |
| $\alpha_2$ | $0.6369^{-1}$ |
| $\alpha_3$ | $0.8451^{-1}$ |
| $\alpha_4$ | $0.9045^{-1}$ |
| $\alpha_5$ | $1.2827^{-1}$ |

**Table 2 Data set calculation results (2).**

| Index | $\beta_2$ | $\beta_3$ | $\beta_4$ | $\beta_5$ |
| --- | --- | --- | --- | --- |
| $\alpha_2$ | $INF^{-1}$ | $0.4439^{-1}$ | $0.5384^{-1}$ | $0.6409^{-1}$ |
| $\alpha_3$ | $0.3655^{-1}$ | $INF^{-1}$ | $0.5819^{-1}$ | $0.7281^{-1}$ |
| $\alpha_4$ | $0.8367^{-1}$ | $0.9391^{-1}$ | $INF^{-1}$ | $1.5871^{-1}$ |
| $\alpha_5$ | $1.4325^{-1}$ | $1.7365^{-1}$ | $1.9325^{-1}$ | $INF^{-1}$ |

calculation indicate that the conditional log-likelihood function in the likelihood function is asymmetric. Thus, it is suitable to be used to identify the directionality of dependence.

# SIMULATION EXPERIMENT ANALYSIS

## Simulation platform construction

In this article, a federated transfer learning of a Bayesian network based on an improved binary algorithm is proposed. By removing the restriction, each client must use $k2$ to learn the Bayesian network algorithm. It eliminates the need for expert knowledge such as node order. Based on the improved NOTEARS dual ascent method, the problem is that the non-$k2$ algorithm will produce a ring after the binary is solved. Therefore, in the experimental part of this article, the effect of network loop removal is firstly verified, compared with the original NOTEARS algorithm applied to the loop removal process; secondly, the performance of the improved binary algorithm is analyzed, and the NOTEARS algorithm and the improved NOTEARS are compared by taking the particle swarm optimization algorithm used by all participants as an example.

The configuration of the experimental platform is displayed in Table 3.

In this article, the NOTEARS algorithm proposed in (*Lee & Kim, 2019*) and the EM algorithm proposed in (*Wang, Ren & Guo, 2022*) are adopted for horizontal comparison. These two comparison methods are the mainstream algorithms at present.

(1) The NOTEARS algorithm is improved, and the loop removal operation is transformed into a constrained optimization problem. Let W be the adjacency matrix of a graph structure with k cycles. There will be non-zero elements on the k-power locus of its adjacency matrix.

(2) EM algorithm, as an iterative algorithm, is a classical method for data completion in the case of missing data (*Saarela, Rohrbeck & Arjas, 2022*). As a single-step iterative algorithm, the main idea is to sample the current node with the knowledge of other nodes, that is:

**Table 3  Experimental platform configuration.**

| Types | Configuration |
| --- | --- |
| CPU | Xeon P8378C 3.6Ghz 38 core/72 threads |
| RAM | 128G DDR4 |
| System | Novell 5.0 |
| Language interpreter | python 3.11 |
| Programming language | Matlab 2022 |
| Toolbox | Full BNT 1.0.7 |

$$T^n = W\sqrt{U\left(\alpha_i\middle|\overrightarrow{\alpha}_j\right)} \tag{6}$$

where: $T^n$ is the *nth* iteration of the data $T$, W is the number of data samples, and $U\left(\alpha_i\middle|\overrightarrow{\alpha}_j\right)$ represents the sampling values of other data except $j$.

## Analysis of experimental results

### Analysis of ring removal effect

The number of participants in federated transfer learning is 100, and the characteristics of each participant are missing with a probability of 0.2. The Bayesian network generated by particle swarm optimization is adopted to analyze the results.

The weight matrix represents the selection of each side by the participants. The larger the weight is, the more votes are obtained. More such sides are expected to remain. The evaluation index is as follows:

$$\delta = \frac{\max\left(\phi_i + \phi_j\right)}{\eta_t} \tag{7}$$

The binary algorithm proposed in this article is compared with the NOTEARS algorithm. The calculation results are displayed in Fig. 4.

Figure 4 presents the results of the algorithm in this article and the NOTEARS algorithm on the binary matrix with ALARM and INSURANCE data sets. It can be seen from Fig. 4 that the number of edges retained by the NOTEARS algorithm in the ring removal process will gradually decrease to zero with the increase of its parameters. Its value requires a heuristic strategy. However, the edges retained by the algorithm in this article will gradually converge to a fixed value with the increase of K. Table 4 indicates the final convergence of the edges of the algorithm in this article in an average of 10 experiments on ALARM, INSURANCE, CHILD and ASIA networks.

The proposed algorithm outperforms NOTEARS algorithm on almost all data sets and outperforms the centralized learning method by a large margin on all given data sets. In this article, we propose a Bayesian network federated transfer structure learning based on the improved binary algorithm, which eliminates the limitation of the fixed learning algorithm in the application of the binary algorithm in Bayesian network federated transfer learning and makes the binary algorithm more universal. The performance of the proposed algorithm is verified by experiments.

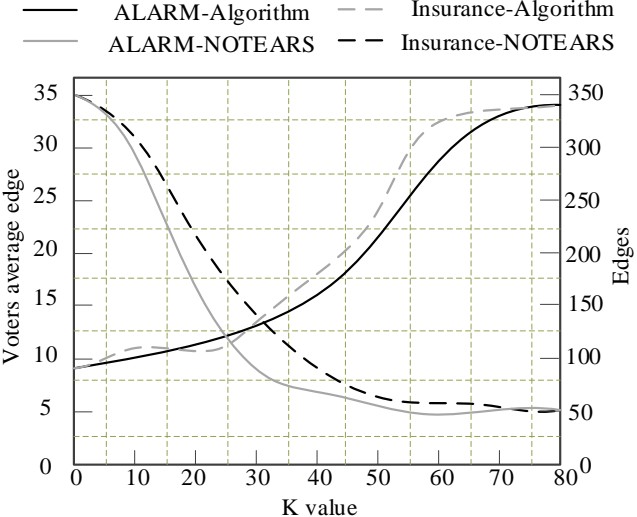

**Figure 4** Variation of different algorithms with the sum.

**Table 4** Convergence data set of each network.

| Index | ASIA | CHILD | INSURANCE | ALARM |
|---|---|---|---|---|
| Number of binary participants | 200 | 200 | 200 | 200 |
| Original sides | 7 | 17 | 25 | 42 |
| Number of edge convergence | 8 | 20 | 34 | 47 |

### Influence of the number of AP clusters on the ensemble

Among the 200 Bayesian regression sub-networks sorted by the validation set, the number of clusters is set to 2-16 by changing the preference P of the AP clustering algorithm. The prediction uncertainty weighting method is adopted to generate conclusions. The traditional NOTEARS algorithm is introduced, and the error of the EM (Expectation-Maximum) algorithm and the proposed algorithm is analyzed in the case of different cluster numbers. Figure 5 presents the clustering error scatter.

In Fig. 5, X-Y is the prediction error accuracy range, where the $Y$-axis "-" value is the unattracted node. The algorithm will eliminate this part, so only the "+" value is taken; O is the best accuracy. O1 is the origin. B is the value radius, and A is the threshold range.

The comparative analysis data is shown in Table 5.

The error dispersion of the prediction model of this algorithm is better than other methods in the comparison of clustering accuracy. The prediction model of the traditional NOTEARS algorithm has better dispersion within the error threshold, but the dispersion convergence outside the threshold is poor. The prediction model of the EM algorithm has the lowest clustering accuracy.

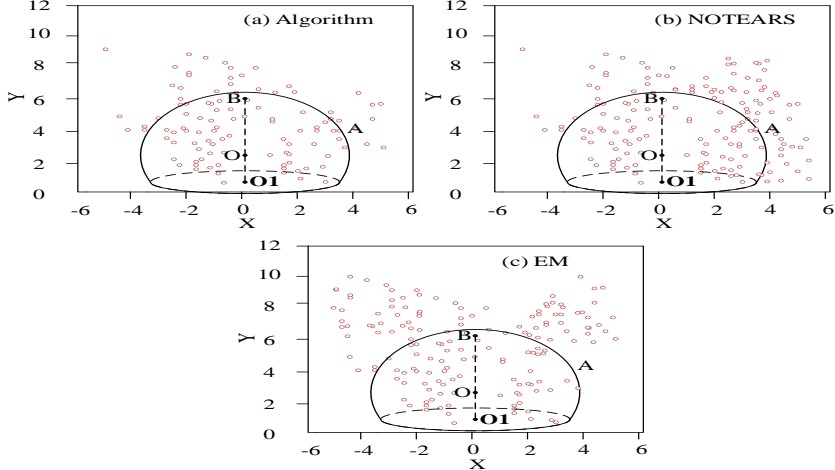

**Figure 5** Comparative analysis of the clustering error dispersion.

**Table 5** Prediction error precision.

| Index | NOTEARS | EM | Algorithm |
|---|---|---|---|
| Average error value | 1.2028 | 1.5637 | 1.9475 |
| Error threshold spread | 0.8501 | 0.8725 | 0.9378 |
| Dispersion convergence | 0.014 | 0.019 | 0.028 |

**Table 6** Comparative analysis of errors of different models.

| Index | NOTEARS | EM | Algorithm |
|---|---|---|---|
| Prediction error | 13.5254 | 15.7479 | 11.5058 |
| Weighted mean | 8.2476 | 9.9631 | 6.2776 |

### Multi-model heterogeneous ensemble learning algorithm result

The strategy of predicting the uncertainty weighting method achieves the minimum prediction error. The error comparison between different models is given in Table 6.

In Table 6, the model in this article achieves the smallest error and the highest accuracy. The accuracy of the results is slightly higher than that of the traditional NOTEARS algorithm and EM algorithm. This fully indicates that even if the prediction accuracy and generalization ability of a single base model is limited, the heterogeneous ensemble learning combined with multiple models can play a complementary role and effectively improve the prediction accuracy and generalization ability of the final ensemble learning model.

### Classifier robustness testing

The Spearman correlation coefficient is defined as the coefficient between the parameters of the rank function. The rank of the function parameter value is selected to calculate the probability distribution and the covariance of the probability distribution. Compared with Pearson's correlation coefficient, it is less affected by outliers, and the neural network

**Table 7** Robustness test analysis.

| Index | NOTEARS | EM | Algorithm |
|---|---|---|---|
| Spearman | 0.0125 | 0.0256 | 0.0579 |
| Pearson | 0.0187 | 0.0319 | 0.0611 |

model that captures the nonlinear relationship is stronger. The calculation method is as follows:

$$\psi_i(ts) = \sum_{i=1}^{\sigma} \frac{\sqrt{(j_i - k_i)}}{(j_i' - k_i')} - 1. \tag{8}$$

In Eq. (8), the Spearman correlation coefficient between the parameters $t$ and $s$ of the function $\psi$, where $\sigma$ represents the number of samples, and $j, k, j', k'$ represents the rank of the $i$th observation of the $j$th function parameter. The test results are shown in Table 7.

Considering that the covariance vector space, namely Pearson correlation coefficient vector space, measures the degree of strict functional connection between two sets of database data series. However, when expressing some data links with obvious random process and correlation relationship, the Spearman correlation coefficient has better performance than Pearson correlation coefficient. Tests indicate that the Spearman correlation coefficient is more robust than Pearson correlation coefficient in capturing positive correlation.

## DISCUSSION

Through the simulation experiment and the performance analysis of the basic structure of the model, the accuracy of the algorithm in this article, the NOTEARS algorithm and the EM algorithm were compared. Then, the causal value was calculated. The accuracy of causal effect data was further improved by simulating various real-time situations with different deep learning models and sample sizes. Finally, a conclusion was drawn to prove that the proposed algorithm was obviously superior to other comparative algorithms in the heterogeneous ensemble learning algorithm of multiple models. In the classifier robust performance test, it was also significantly better than the traditional classification algorithm.

## CONCLUSION

In this article, a heuristic search strategy was adopted to relax the independence assumption of SPODE by exploring the high-order dependencies between attributes.

(1) In the process of structure extension, it not only realizes the recognition of the dependency relationship between attributes, but also takes into account the direction recognition of the dependency relationship, and identifies the parent–child relationship between dependencies.

(2) In the case of large-scale data, the advantages of classification accuracy and bias are more significant.

(3) According to the experimental analysis, the binary classification model is applied to the network data set. The performance of the model is evaluated by comparing the differences in classification accuracy of each model. The results verify that the Bayesian network model based on the binary classification optimization algorithm is superior to the comparison method in all test results, and can be used in a variety of classifier models, with a certain degree of universality. At the same time, this method can improve the classification accuracy of many traditional classifiers, so it can play an important role in practical applications.

The logarithmic likelihood function can take into account both the dependence and the directional dependence, so it is a very effective measure. It is also valuable to use the log-likelihood function in other improvements of Bayesian network classifiers, such as attribute weighting and model weighting. Therefore, the research on the selection and weighting of Bayesian network classifiers using the log-likelihood function will be the next research direction and focus.

### Funding
This work was supported by the Guangdong Provincial Higher Education Teaching Reform Project (No. 20191206). The funders had no role in study design, data collection and analysis, decision to publish, or preparation of the manuscript.

### Grant Disclosures
The following grant information was disclosed by the author:
The Guangdong Provincial Higher Education Teaching Reform Project: 20191206.

### Competing Interests
The author declares that there are no competing interests.

### Author Contributions
- Yongna Yao conceived and designed the experiments, performed the experiments, analyzed the data, performed the computation work, prepared figures and/or tables, authored or reviewed drafts of the article, and approved the final draft.

### Data Availability
   The code and data are available in the Supplemental Files.

### Supplemental Information
Supplemental information for this article can be found online at http://dx.doi.org/10.7717/peerj-cs.1466#supplemental-information.

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
