# Peer review of "Bayesian network model structure based on binary evolutionary algorithm"

_PeerJ Computer Science, doi:10.7717/peerj-cs.1466_

## Round 0.1 · original submission · Minor Revisions

Based on two reviewers' comments and my own reading, I think the paper generally shows an interesting idea and some novelty has been demonstrated. However, additional efforts are still needed to clarify a few things as listed.

Reviewer 1 ·

Basic reporting

In this paper, Bayesian network structure learning, analysis of the shortcomings of traditional algorithms, and binary evolutionary algorithm is applied to the randomized algorithm to generate the initial population. The discussion of Bayesian reasoning in this paper has very important theoretical and practical significance, which not only provides a direction for people to mine and understand probability information but also has a certain guiding role for people to learn effectively and make decisions based on probability theory. This study is interesting however there are some drawbacks that the authors should address them to improve their study.

Experimental design

no comment

Validity of the findings

no comment

Additional comments

comments:
1.In the introduction, the research gap is lacked to state clearly the object of study especially the selection of proposed method based on Bayesian network structure model.
2.In Figure 3, the network model operation flow, the block “Binary clustering algorithm”, checking the condition “Judfe the current stage” will have two cases “yes” and “no”. However, both two cases finally lead to the update the optimal solution. So, what is the difference between two cases? This reviewer could not see the the clustering results from this algorithm.
3.In table 5, there are two parameters: prediction error and weighted mean have been discussed with two other algorithms. It is inadequate to confirm that this proposed method is the most priority than other previous methods. The authors should enlarge and discuss more some other research results based on the Bayesian network structure into the data clustering analysis.
4.There are further comments which have been added in the manuscript.
5.There are some grammatical mistakes in English which have been found in this paper.

Cite this review as

Reviewer 2 ·

Basic reporting

The idea is good but some issues need to be covered.
1. Comparison table should be added.
2. Pseudo code is compulsory to add.
3. Latest references must be used. no longer than 5 years. reference 32 is 2004.
4. Author discuss error in the whole paper. but accuracy didn't discuss in the whole paper. Please discuss the exact accuracy of your proposed work.
5. Could you provide all the accuracy matrices for the classification model in the results section?
6. It would be helpful to include information on the dataset used for model development and explain the significant characteristics of the data.
7. Could you please list the key differences between NS-DST and other models from the literature?
8. In Figure 4, what does the k value represent and what is its significance?

Experimental design

NA

Validity of the findings

NA

Additional comments

NA

Cite this review as

---

## Round 0.2 · accepted · Accept

With both reviewers' and my own reading, I think the final manuscript is in a good shape and ready to publish.

Reviewer 1 ·

Basic reporting

The author answered all the questions and the quality of the paper was greatly improved.

Experimental design

no comment

Validity of the findings

no comment

Additional comments

no comment

Cite this review as

Reviewer 2 ·

Basic reporting

NA

Experimental design

NA

Validity of the findings

NA

Additional comments

NA

Cite this review as